# Effects of Porous Supports in Thin-Film Composite Membranes on CO_2_ Separation Performances

**DOI:** 10.3390/membranes13030359

**Published:** 2023-03-21

**Authors:** Hongfang Guo, Wenqi Xu, Jing Wei, Yulei Ma, Zikang Qin, Zhongde Dai, Jing Deng, Liyuan Deng

**Affiliations:** 1College of Architecture and Environment, Sichuan University, Chengdu 610065, China; 2Carbon Neutral Technology Innovation Center of Sichuan, Chengdu 610065, China; 3National Engineering Research Centre for Flue Gas Desulfurization, Chengdu 610065, China; 4School of Carbon Neutrality Future Technology, Sichuan University, Chengdu 610065, China; 5Yibin Institute of Industrial Technology, Sichuan University, Yibin 644000, China; 6Department of Chemical Engineering, Norwegian University of Science and Technology (NTNU), 7491 Trondheim, Norway; 7ALTR FLTR Inc., Phoenix, AZ 85034, USA

**Keywords:** porous support, thin-film composite, membranes, humid condition, CO_2_ separation

## Abstract

Despite numerous publications on membrane materials and the fabrication of thin-film composite (TFC) membranes for CO_2_ separation in recent decades, the effects of porous supports on TFC membrane performance have rarely been reported, especially when humid conditions are concerned. In this work, six commonly used porous supports were investigated to study their effects on membrane morphology and the gas transport properties of TFC membranes. Two common membrane materials, Pebax and poly(vinyl alcohol) (PVA), were employed as selective layers to make sample membranes. The fabricated TFC membranes were tested under humid conditions, and the effect of water vapor on gas permeation in the supports was studied. The experiments showed that all membranes exhibited notably different performances under dry or humid conditions. For polyacrylonitrile (PAN) and poly(ether sulfones) (PESF) membranes, the water vapor easily condenses in the pores of these supports, thus sharply increasing the mass transfer resistance. The effect of water vapor is less in the case of polyvinylidene difluoride (PVDF) and polysulfone (PSF), showing better long-term stability. Porous supports significantly contribute to the overall mass transfer resistance. The presence of water vapor worsens the mass transfer in the porous support due to the pore condensation and support material swelling. The membrane fabrication condition must be optimized to avoid pore condensation and maintain good separation performance.

## 1. Introduction

Membranes are well known as promising alternatives to conventional separation technologies (e.g., distillation, absorption, and pressure swing adsorption) due to their simple operation, being easy to scale-up, low energy cost, and easy maintenance [1,2,3]. In recent decades, membranes have been used in many fields, such as seawater desalination, wastewater treatment, and gas separation (e.g., O_2_/N_2_ separation, H_2_ recovery, natural gas sweetening, and more recently, CO_2_ capture) [4,5,6]. In the case of gas separation membranes, particularly CO_2_ separation membranes, numerous materials have been developed in recent years, but most reports are on intrinsic gas separation properties obtained from single gas permeation tests using freestanding thick films (50~100 μm) [7]. 

However, asymmetric membranes, such as multilayer composites, are used for real industrial applications to maximize their separation efficiency. A multilayer composite membrane typically consists of a thin selective layer mechanically supported by a microporous substrate [8]. In many cases, a gutter layer and a protective layer are also employed. To date, tremendous efforts have been made to develop new membrane materials with high separation performances for CO_2_ separation [9,10,11,12], but much less has been done to understand the influences of the porous support or to develop/improve new porous supports suitable for TFC membranes. 

The importance of the porous supports on liquid separation membranes has received significant attention from researchers. For instance, Ghosh et al. reported the impacts of the support membranes’ structure and chemistry on polyamide–polysulfone interfacial composite membranes in 2009 [13]; in another study, Ramon et al. investigated the effects of the support membrane pore size and porosity on diffusive transport through composite membranes using theoretical models [14,15]. In a later report published in 2013, Misdan et al. studied the effect of polysulfone substrate characteristics on the fabrication of polyamide membranes [16]. All these papers attracted significant attention from among researches and were widely cited.

The effects of porous supports employed in TFC membranes for gas separation receive much less attention, with only a few papers found in the literature [17,18,19,20,21] and much fewer citations. Beuscher et al. published a series of papers focusing on analyzing the influence of porous supports [22]. They found out that the transport of gas mixtures in porous membrane support is a complex combination of Knudsen diffusion, continuum diffusion, and viscous flow. The resistance of the support layer normally dominates the fast gas (e.g., VOC) permeation. Later, Liu et al. studied the effect of porous support on the overall gas transport properties in TFC membranes [17]. However, they found out that porous supports contribute as much as 75–98% of the total permeation resistance, resulting in a significant reduction in both gas permeance and selectivity. Deng and co-workers studied the gas mass transfer in a silicone rubber-PSF TFC membrane in 2001 [19]. In another study, Lin et al. employed a TFC membrane for natural gas dehydration application; he also found out that the porous support contributes much bigger resistance to the fast transport species (i.e., water vapor) [18]. Their work clearly shows that future work should be focused not only on membrane materials with superior gas permeability but also fabricating new porous support with lower mass transfer resistance.

In recent decades, a large number of CO_2_ separation membranes have been reported [23,24,25,26], including facilitated transport membranes and other CO_2_-philic membranes. The fabrication of TFC membranes to optimize membranes for high performance in real industrial applications has also been extensively studied. However, the effects of porous supports in TFC membranes have rarely been reported. As most CO_2_-containing streams include water vapor, and water is involved in CO_2_ transport mechanism, water vapor is critical in membranes for CO_2_ separation. Water vapor could not only contribute to the reversible reaction involved in the facilitated transport mechanism and swell the selective layer membrane materials [27,28]. However, these also significantly affect the mass transfer in porous supports. Therefore, in the present study, six different types of the most commonly used and commercially available flat sheet supports were selected to fabricate TFC membranes and study their effects on the overall separation performance. Two representative polymers, Pebax 1657 and poly(vinyl alcohol) (PVA), were selected as the coating layer materials in this study as they are among the most reported materials in CO_2_ separation membranes. The gas separation performances of these membranes were studied under fully humid conditions to understand the gas transport and optimize the stability of the selected porous supports. Additionally, the porous support morphology and the surface hydrophilicity/hydrophobicity were analyzed using SEM and the contact angle (CA), respectively.

## 2. Materials and Methods

### 2.1. Materials

Polysulfone (PSF) membranes with a molecular weight cut-off (MWCO) of 20k and 50k, and polyethersulfone (PESF) with MWCO of 30k and 50k were purchased from Alpha Laval, Danmark. Polyacrylonitrile (PAN) was kindly provided by Fujifilm. Polyvinylidene difluoride (PVDF) with an MWCO of 30k was also purchased from the same company. Pebax 1657 pellets were ordered from Arkema, Colombes, France. Poly(vinyl alcohol) (PVA) (M_n_ 85–124 g/mol) and absolute ethanol (EtOH) were ordered from Sigma, Schnelldorf, Germany. All the chemicals were used without further treatment.

### 2.2. Membrane Preparation

Pebax 1657 and PVA were selected as selective layer materials. 1 wt.% PVA solution was prepared by dissolving the PVA in deionized (DI) water at ~80 °C for 4 h. 1 wt.% Pebax 1657 solution was prepared by dissolving the polymer in the EtOH/H_2_O (70/30 vol.%) mixture at 80 °C with reflux for 4 h. All the porous support samples were washed in tap water for 2 h, and then in DI water overnight before use. The selective layer was coated via a bar-coating machine (Elcometer 4340, Elcometer Instruments GmbH, Arlen (Baden-Wurttemberg) Baden-Wurttemberg, Germany) and knife casting (KTQ-100, SN:20168). With a coating wet gap of 100 μm and a moving speed of 5 (mm/s). Because the contribution of a gutter layer is relatively small compared to the porous support and the selective layer, in this study, the gutter layer was not used.

### 2.3. Characterization

Membrane morphology was investigated using a scanning electron microscope (SEM, TM3030 tabletop microscope, Hitachi High Technologies America, Inc., Schaumburg, IL, USA). All samples were sputter-coated with conducting gold before the SEM test to avoid electrical charging. 

Contact angle (Attension tensiometer, Biolin Scientific, Göteborg, Sweden) was carried out to check the membrane hydrophilicity. DI water was used as the liquid phase and the CA was determined from the average value of three measurements. 

A gas separation test was conducted using a mixed-gas permeation setup as described in detail elsewhere [29,30]. A CO_2_/N_2_ (10/90 *v*/*v*) gas mixture was used as the feed gas, whereas pure CH_4_ was used as the sweep gas. The humidity of both the feed and sweep streams was set to 100%. The feed-side pressure was controlled by a back-pressure regulator (El-Press series, Bronkhorst, Ruurlo, The Netherlands). Pressures were monitored (Wika, S-10) and held constant at 2.0 bar on the feed side and up to 1.05 bar on the sweep side for all the experiments. The compositions of retentate and permeate streams exiting the membrane module were monitored by a calibrated gas chromatograph (490 Micro GC, Agilent, Santa Clara, CA, USA) throughout the test. Each test continued for at least 6 h to ensure a steady state.

The permeance (*P_i_*) of the *i*th penetrant species was measured by Equation (1)
(1)$$P_{m,i}=\frac{N_{perm}\left(1-y_{H_{2}O}\right)y_{i}}{A\left(\langle p_{i,feed},\;p_{i,ret}\rangle -p_{i,perm}\right)}$$
where *N_perm_* is the total permeate flow measured with a bubble flow meter, *y_H_*__2_*O*_ is the molar fraction of water in the permeate flow (calculated according to the relative humidity (RH) value and the vapor pressure at the given temperature), *y_i_* is the molar fraction of the species of interest in the permeate, and *p_i,feed_*, *p_i,ret_*, and *p_i,perm_* identify the partial pressures of the *i*th species in the feed, retentate and permeate, respectively. The separation factor ($\alpha_{i/j}=\frac{y_{i}/x_{i}}{y_{j}/x_{j}}$) was applied to the mixed-gas permeation tests. 

## 3. Results and Discussion

### 3.1. Porous Support

#### 3.1.1. Morphology Study

The morphologies of the six supports selected were studied using SEM, and their cross-sections are shown in Figure 1. The PAN support and PSF 50k membrane present a sponge-type structure, while the PSF 20k, PESF 30k, and PESF 50k membrane show a finger-type porous structure. The PVDF 30k membrane shows a different structure compared to the other five samples. It has sponge type pores at the bottom side of the support close to the non-woven fabric, but big pores with a diameter of 1~5 μm present just under the selective layer (~1 μm). The different structures of these materials may come from a different membrane fabrication process.

As discussed previously, the support plays a significant role in the overall mass transfer in the TFC membranes. Therefore, in principle, a support with lower thickness and higher porosity is preferred for fabricating membranes. The thicknesses of the six porous supports were also studied using SEM and summarized in Table 1. As shown in Table 1, the PSF membrane has the highest total thickness in all the selected membranes, which is approximately 150 μm. The PESF membrane and PVDF membrane exhibit a similar total thickness of ~80 μm. The thinnest membrane comes from the PAN support from Fujifilm, which shows a value of ~40 μm.

The support surface is another critical parameter that needs to be considered in TFC membrane fabrication. A smooth surface is a prerequisite for fabricating membranes with an ultrathin selective layer (<1 μm). The surface images of the as-received membranes are shown in Figure 2. As can be seen, unknown particles within the micrometer range are easily found on the surfaces of all membranes. In addition, defects with a large dimension (a few μm in width and dozens of μm in length) can be found on both PSF membranes. The overall smoothness of these membranes follows the trend of PAN/PVDF > PESF 50k > PESF 30k > PSF 20k >= PSF 50k.

As indicated by the supplier, the PSF membranes contained pore-protecting agents that needed to be removed before use. The recommended procedure involved washing the membrane with tap water at 45 °C for 2 h, followed by DI water for one night. Other membranes do not need to remove additives, but micro-sized particles were found on all samples; thus, all the supports were washed using the same procedure. The surface images of the thoroughly washed membranes are shown in Figure 3. Compared to the as-received membranes (Figure 2), most of the particles on the surface were removed from the membrane surfaces. 

On the other hand, the two PSF membranes show quite interesting results: after washing by the suggested protocol, it was found that plenty of unknown particles with a typical crystal appearance showed up on the membrane surface, as shown in Figure 4. It may come from the precipitation of the pore-protecting agent used by the supplier with insufficient wash. With this type of membrane surface, it is impossible to obtain a defect-free TFC membrane. To overcome this, the wash period was extended from 2 h (1 h tap water and 1 h DI water) to overnight (2 h tap water and overnight DI water), most of these particles can be removed.

#### 3.1.2. Contact Angle(CA) Study

In the TFC membrane fabrication process, it is well known that the surface hydrophilicity/hydrophobicity has a dominating effect on the final membrane thickness and morphology. CA is commonly used to indicate that a surface is hydrophilic (CA < 90°) or hydrophobic (CA > 90°). In the present study, the CA value of the six selected supports was measured using DI water. The CA test was carried out using the cleaned supports. All the CA tests were carried out at least three times, and the average values are presented in Figure 5. 

As shown in Figure 5, surprisingly, the PVDF membrane shows the lowest CA (approximately 40°) among all samples. Due to the high fluoride content in the polymer, the PVDF polymer normally shows a CA value in the range of 70~85°, depending on the pore size/porosity and surface morphology [31]. However, the CA value of PVDF can be significantly lower if hydrophilic surface modification is applied [31,32]. In the present study, the CA value for PVDF is much lower than the literature data, possibly due to the hydrophilic surface modification by the supplier. PAN shows a slightly higher CA value (50–60°) compared to the PVDF sample. PSF and PESF supports exhibit even higher CA values in the selected samples. Although the reasons are still unclear, for both PSF and PESF support materials, it is found that the supports of higher MWCO exhibit a higher CA value.

#### 3.1.3. Gas Permeation Properties under Humid Conditions

Many CO_2_ facilitated transport membranes involve water vapor in the facilitated transport mechanism. However, the water vapor may condense in the pores of the porous support. The condensed water vapor in the pores gradually increases the overall mass transfer resistance of the membranes, and consequently reduces the membrane separation efficiency. Therefore, the CO_2_ permeation performance of these support membranes was studied under fully humid conditions, and the results are shown in Figure 6.

As shown in Figure 6, the support membranes exhibit rather high CO_2_ permeances under dry state conditions (the first 1–2 min); however, a significant reduction in the CO_2_ permeance can be found in five of the six selected supports. The only exception is the PVDF support. It shows the CO_2_ permeance of ~1000 GPU at the dry state and this value remains under humid conditions during the test period (~1500 min). The PAN support has the highest starting value, which is over 10,000 GPU; however, this value dramatically reduces under the humid condition in a very short time. After 500 min, the CO_2_ permeance reduced to ~20 GPU. Similar phenomena are observed for the two PSF and PESF supports. Even though the starting point of these supports is reasonably high, they gradually reduce to lower than 100 GPU. Based on the resistance in the series model in mass transfer, the overall gas transport resistance is dominated by the selective layer. The total gas permeance of a composite membrane coated on the supports is always expected to be lower than this value, which means that using these supports to fabricate TFC membranes may not result in high permeation for membrane separation when involving water vapor. 

### 3.2. Membrane Coated with Pebax 1657

#### 3.2.1. Coating Condition Selection

It is well known that the selective layer thickness will result in differences in both gas permeance and selectivity [8,9,33] in facilitated transport membranes or when condensable vapor is involved in the separation system. As varying coating solution concentrations result in different membrane thicknesses, the effect of the Pebax solution concentration on the final membrane performances was studied, as shown in Figure 7. 

As shown in Figure 7, increasing the coating solution concentration from 0.25 wt.% to 4 wt.% resulted in a notable decrease in the CO_2_ permeance. Starting from membranes of 0.25 wt.% Pebax solution, the CO_2_ permeance of up to 1150 GPU was obtained; however, the CO_2_/N_2_ selectivity was as low as ~1.5, denoting that the Pebax layer did not fully cover the PAN support. Increasing the concentration to 0.5 wt.% resulted in a sharp decrease in CO_2_ permeance, of approximately 400 GPU, but the CO_2_/N_2_ selectivity was increased to 12. The selectivity obtained from this membrane was still lower than the intrinsic selectivity of Pebax 1657; thus, the coating solution concentration was further increased. The trend for CO_2_ permeance and CO_2_/N_2_ selectivity continues. Therefore, 1.0 wt.% Pebax solution was selected as the optimized coating concentration for further study. 

The long-term stability is another critical factor that needs to be considered for membrane separation. The membrane fabricated via 1.0 wt.% coating concentration on PAN support was selected to study the stability. As shown in Figure 8, the membrane shows a relatively stable separation performance during the tested period. Only a slight reduction can be found for the CO_2_ permeance; at the same time, a slight increase in CO_2_/N_2_ selectivity was also observed, and possibly, these small changes come from the fact that the feed pressure compacted the selective layer. In general, the overall separation performances of the Pebax membranes are good.

#### 3.2.2. Morphology Study

The morphology of the Pebax coated on different supports was studied using SEM (Figure 9).

As shown in Figure 9, it is found that, even though the coating conditions are the same for all supports, notable differences in the selective layer thickness were obtained. The PESF support has the highest selective layer thickness value among the six selected supports, i.e., 0.81~1.12 μm. The selective layer obtained from the other four selected supports is thinner, within a range of 0.15–0.32 μm. Details of the selective layer thickness are listed in Table 2.

The membrane surface is also presented in the insets of Figure 9. It is found out that those obtained membranes show a rather similar structure. Some dots can be found on the surfaces of all the samples, which is a typical Pebax membrane morphology due to microphase separation that has been reported in different works on the subject [34].

As we know, coating parameters affect the morphology of the TFC membranes, such as the coating solution concentration, the casting knife moving speed, and the wet casting gap [35]. Interestingly, this work found that, even when using the same coating solution and casting parameters, different selective layer thicknesses were obtained on different supports. The intrinsic material properties and the surface morphologies of the supports present a strong effect on the final membrane thickness. Thus, the casting condition for different supports should be optimized case by case. 

#### 3.2.3. Contact Angle

Due to the fact that the six selected supports have a large difference in chemical structure and morphology, significant differences in the contact angle were observed (Figure 5). The Pebax layer in the TFC membrane determines the contact angle after coating with the 1.0 wt.% Pebax solution, showing a different contact angle to that of the pristine support. 

After coating a layer of Pebax onto the PAN support, the CA value of ~50° was obtained, which is approximately 10° lower than the pristine PAN support, as shown in Figure 10. On the other hand, after the Pebax coating, the CA value of the Pebax/PVDF slightly increased to ~60°, which means that the Pebax dominates the CA value in the Pebax/PVDF membrane, which is close to the CA value of pristine Pebax 1657 (60~65°). In terms of PSF supports, the CA value of both PSF 20k and PSF 50k samples were reduced from hydrophobic (85°, and 95°, respectively) to hydrophilic (<=50°, and 70°, respectively), and it supports the fact that the PSF 50k sample still holds the higher CA value. In the case of PESF supports, similar phenomena were found, notably that the CA value for both samples decreased to a hydrophilic state. The overall CA value for all the coated support is generally in a similar range, i.e., 40~60°.

#### 3.2.4. Gas Permeation under Humid Conditions

Gas permeation of the TFC Pebax membranes coated onto the six different supports was tested under fully humid conditions at both the feed and permeate sides, as shown in Figure 11.

As can be seen from Figure 11, except for the membranes cast on the supports of PVDF and PSF 50k, those of the other four supports experienced a reduction in CO_2_ permeance over time. The membrane cast on the PAN support started with a rather high CO_2_ permeance (~500 GPU). However, the permeance decreased sharply in a short period to a low value (~20 GPU). It is well accepted that increasing the RH value will enhance the Pebax gas permeability [36]. For instance, in our previous study, the CO_2_ permeability was greatly improved as the RH value increased [27,37]. However, in the present study, the CO_2_ permeance of the Pebax/PAN membrane was sharply reduced during humidification, denoting that the PAN support dominated the gas transport in the TFC membrane, rather than the Pebax layer. The water vapor condensed in the PAN pores and swelled the PAN polymer, which resulted in an increase in the mass transfer resistance of the support layer, leading to a sharp reduction in the gas permeation rate. 

Surprisingly, even though the PVDF support shows a relatively low CA, the Pebax/PVDF membrane exhibited much better stability under humid conditions. Although the CO_2_ permeance was in the low region (~40 GPU), no significant reduction can be found in the whole gas permeation test period (~2000 min). This finding clearly shows that water vapor has a limited effect on the Pebax/PVDF membrane, and a possible explanation can be that it is more difficult for water vapor to condense along the surface of the pores in hydrophobic PVDF support.

It is also interesting that the membranes coated on supports with the same materials but different MWCO supports do not behave in the same manner. A membrane coated on PSF 50k support shows higher CO_2_ permeance compared to the PSF 20k samples: the CO_2_ permeance of over 100 GPU was maintained in a time of ~1500 min. On the other hand, the PESF 30k delivers a more stable and permeable TFC membrane than the analog with higher MWCO. These may be associated with the different morphology and surface properties of the porous supports. 

### 3.3. Membrane Coated with PVA

#### 3.3.1. Morphology Study

The morphology of the PVA TFC membrane was studied with SEM and the results are shown in Figure 12.

Similar to the Pebax case, the PVA membrane fabricated on the PESF supports shows the highest selective layer thickness, ranging from 0.35 to 0.69 μm, followed by the PAN support, which is approximately 0.33 μm. PSF supports exhibited the lowest thickness. The detailed selective layer thicknesses can be found in Table 3.

As shown in Table 3, the thickness behavior is quite similar to the Pebax case, demonstrating that the porous support has a significant effect on the selective layer thickness. The optimization of the selective layer for different supports should be optimized individually. The membrane surface is smooth for all samples and similar to other reports using PVA as a selective layer [38].

#### 3.3.2. Contact Angle

The CA of the PVA coated on various supports was tested again and the results are shown in Figure 13. The CA value of the self-standing PVA was also tested and a value of ~60° was obtained [39].

Compared to the corresponding supports, the coating of PVA onto these supports increases the hydrophilicity of these supports, which are rather similar to the literature reports [40]. The CA value gradually reduces over time, demonstrating that the water quickly swelled the PVA layer. For PVDF and PAN supports, the CA value change was not significant, but the slope of reduction is much steeper after PVA coating. The PSF supports have the biggest reduction in CA value, and the coated PSF/PVA membrane exhibited a CA value close to the intrinsic value of the PVA material, denoting that the PVA dominates the CA value. With regard to the PESF sample, even though the neat PESF 30k and 50k show slight differences in the CA value, after being coated with PVA, both membranes show a CA value of ~80°, and also reduce quickly over time. It is well known that PVA has high water permeability [39]; therefore, in the case water vapor condensed on the PVA surface, it will penetrate through the selective layer and condense in the porous supports.

#### 3.3.3. Gas Permeation at Humid State

The gas permeance of the PVA TFC membrane fabricated on different supports was tested using 100% RH conditions, and the results are shown in Figure 14.

As shown in Figure 14 for the PVA/PAN membrane, a gradual reduction in the CO_2_ permeance can be seen. Starting from ~120 GPU, the PVA/PAN membrane CO_2_ permeance gradually reduced to 69 GPU in ~10 h (600 min). The good news is that the membrane gas permeance stabilized and no further reduction was observed. Compared to the gas permeance results from the neat PAN support, which shows a CO_2_ permeance of ~20 GPU after it became fully swollen due to water vapor, the PVA layer seems to play a positive role in preventing the porous supports becoming fully swollen.

In the case of PVDF support, the CO_2_ permeance of ~50 GPU was obtained, which was quite similar to the Pebax/PVDF membrane. Possibly, the relatively low CA of the PVDF membrane makes the support rather easy to penetrate by the water or water/ethanol coating solution. The polymer in the penetrated coating solution blocked the PVDF pores and reduced the overall gas permeance. In a future study, a pore-filling agent (e.g., AF 72) can be used to reduce the possible penetration and increase the membrane separation performances.

Considering the PSF supports, even though the PSF 50k has a relatively bigger molecular weight cut off, the stability of the PVA/PSF 50k is better than the PVA/PSF 20k. Starting with the initial CO_2_ permeance of 271 GPU, the CO_2_ permeance of the PVA/PSF 50k membrane reduced to ~70 GPU in only half an hour. This value stabilized and remained for 1000 min. On the other hand, the PVA/PSF 20k membrane has a rather bad stability test; it even started with similar CO_2_ permeance at the beginning (267 GPU). It reduced to only 3 GPU in 200 min; thus, the test was discontinued.

The PVA/PESF membrane exhibited even worse results as shown in Figure 14 For both cases, they started with the CO_2_ permeance of ~100 GPU, but both PESF 30k and PESF 50 reduced to ~20 GPU in less than 20 min. 

## 4. Conclusions

In the current work, the six most commonly used porous supports were selected to fabricate TFC membranes and study the effects of different support substrates on their CO_2_ separation performance under humid conditions. Two hydrophilic membrane materials, Pebax and PVA, were employed to cast as selective layers on the supports. The porous supports and fabricated TFC membranes were characterized using various techniques, including CA, SEM, and gas permeation test under fully humid conditions. The key findings of the current study are:

(1) For the six commercially available porous supports, washing the supports before coating is necessary to prevent particles aggregating on the surfaces of the support, which causes the formation of defects in the thin coating layer.

(2) The CA of the porous supports not only depends on the intrinsic properties of the polymer but also on the membrane surface morphology.

(3) In TFC membrane fabrication processes, apart from the coating parameters (e.g., coating speed, solution concentration, and wet coating gap), the material properties and surface morphology of the porous supports also have critical impacts on the morphology and separation performance of the final TFC membranes.

(4) Among the six supports selected in the current work and operated under fully humid conditions, unfortunately, for PAN and PESF membranes, water vapor could easily condense in the pores of these supports and sharply increase the mass transfer resistance. In the case of PVDF and PSF 50k, the effect of water vapor is not that obvious, which shows relatively better long-term stability. It is also worth mentioning that the selective layer, the gutter layer, and the operation conditions (e.g., temperature, RH value, and pressure) may change the pore condensation behavior, leading to rather different gas separation performances.

Porous supports significantly contribute to the mass transfer resistance in a TFC membrane for gas separation, especially under humid state conditions. More research work should be carried out on developing or optimizing supports with low mass transfer resistance. In addition, the operation parameter of a facilitated transport membrane must be optimized to avoid the pore condensation in the support to reduce the mass transfer resistance and increase the overall separation efficiency.

## Figures and Tables

**Figure 1 membranes-13-00359-f001:**
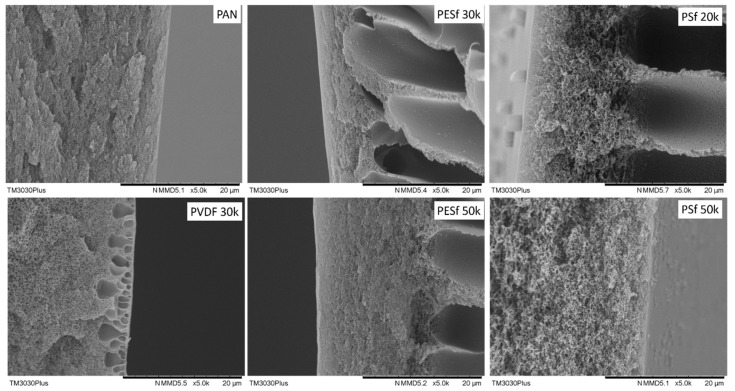
Cross-section of the 6 porous supports selected.

**Figure 2 membranes-13-00359-f002:**
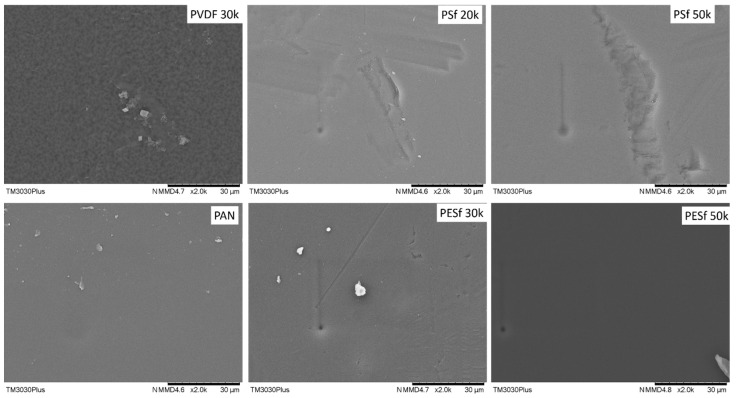
Surface morphology of the 6 porous supports selected—unwashed.

**Figure 3 membranes-13-00359-f003:**
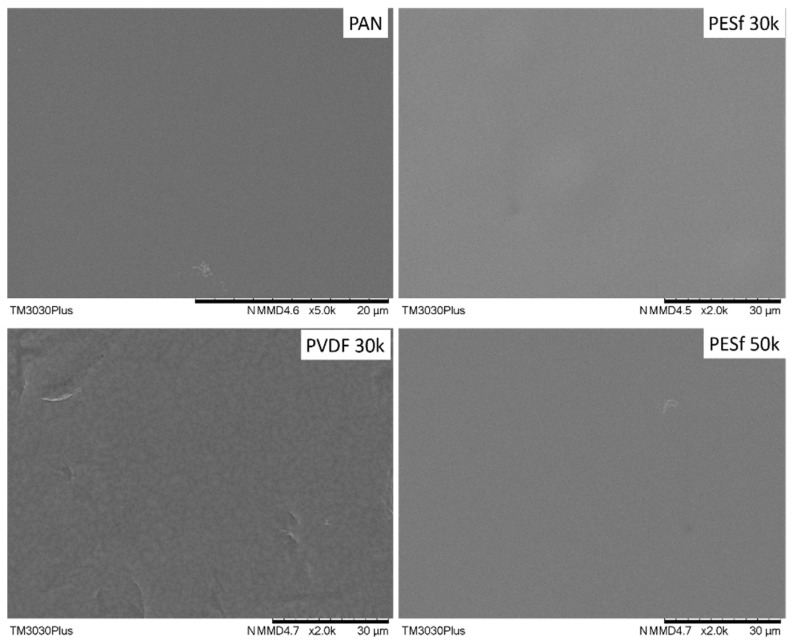
Surface morphology of the PAN, PESF 30k, PVDF 30k, and PESF 50k porous supports after being washed with tape water and DI water.

**Figure 4 membranes-13-00359-f004:**
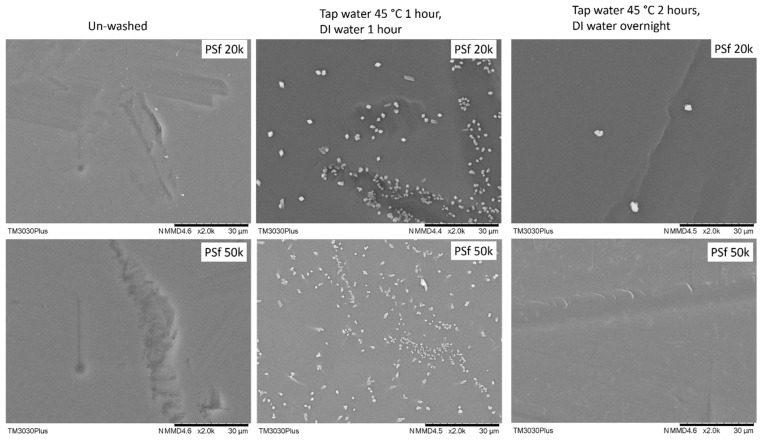
Surface morphology of 2 PSF porous supports washed by different procedures.

**Figure 5 membranes-13-00359-f005:**
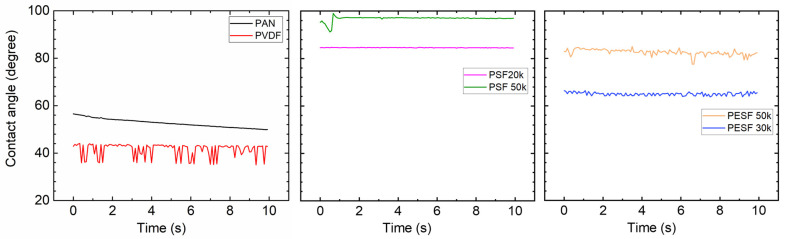
Contact angle of the 6 porous supports selected.

**Figure 6 membranes-13-00359-f006:**
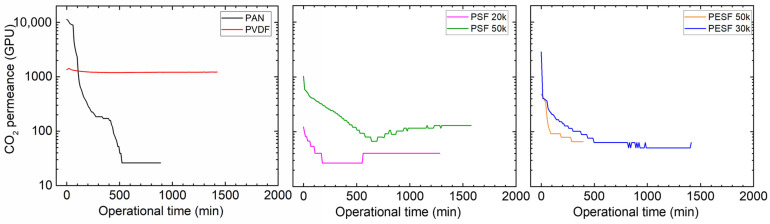
CO_2_ permeation of the 6 porous supports selected at 100% RH, with a feed pressure of 2 bar at 35 °C.

**Figure 7 membranes-13-00359-f007:**
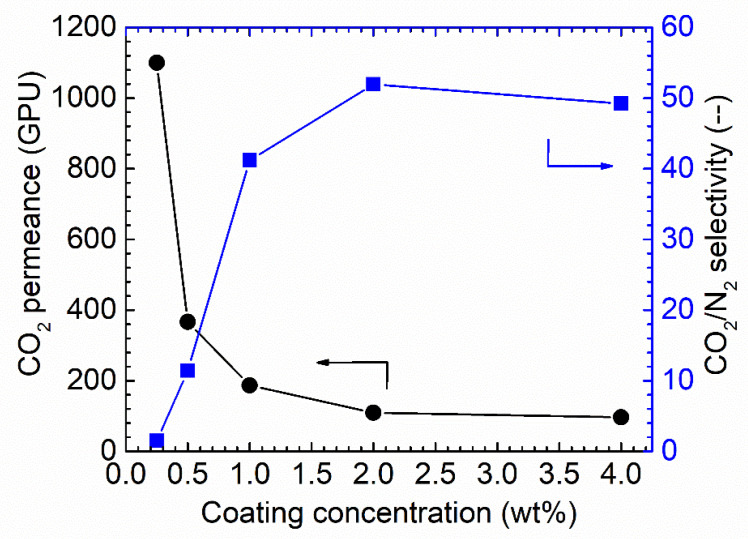
CO_2_ separation performances under the humid conditions for the Pebax/PAN membrane-coated via various coating concentrations, 100% RH, feed pressure of 2 bar at 35 °C.

**Figure 8 membranes-13-00359-f008:**
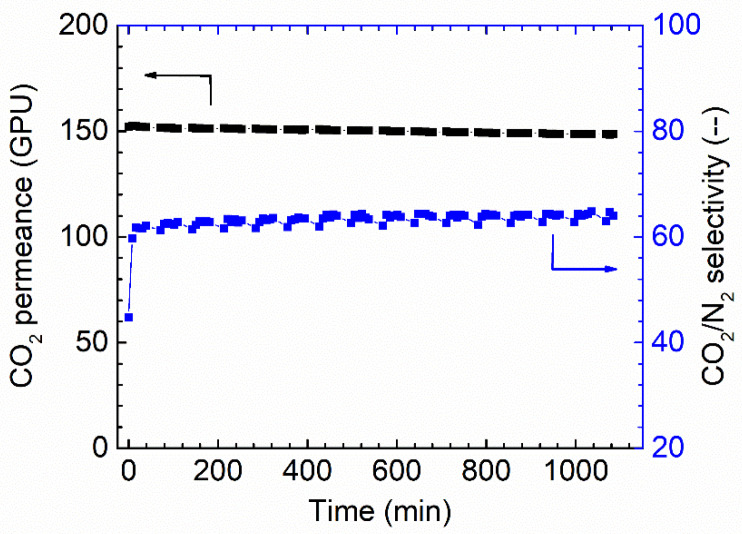
Stability performances of Pebax membrane in the dry state (1 wt.%), 0% RH, feed pressure of 2 bar at 35 °C.

**Figure 9 membranes-13-00359-f009:**
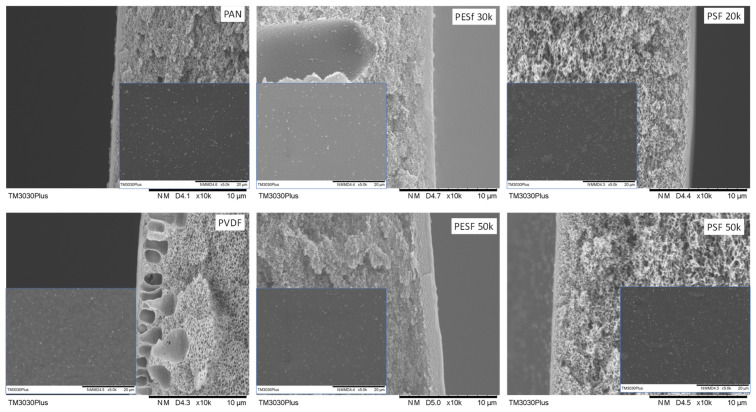
Cross-section and surface (inset) images of the Pebax TFC membranes coated on different supports.

**Figure 10 membranes-13-00359-f010:**
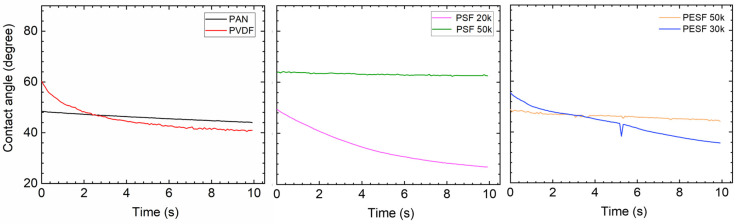
The CA of the 6 selected porous support coated with Pebax 1657.

**Figure 11 membranes-13-00359-f011:**
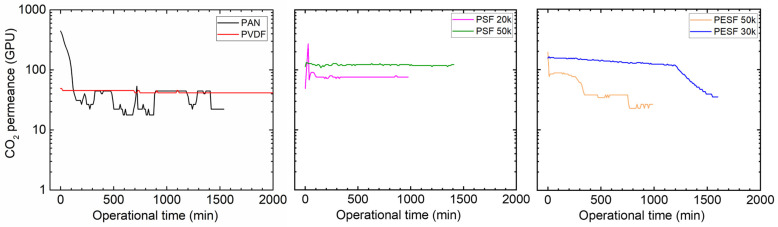
Gas permeation of Pebax TFC membranes coated on various supports with a feed pressure of 2 bar at 35 °C, 100%RH.

**Figure 12 membranes-13-00359-f012:**
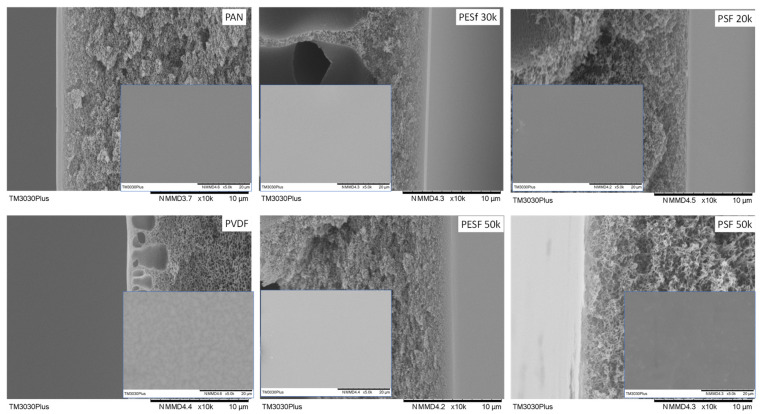
Membrane morphology of PVA TFC membranes coated on selected supports.

**Figure 13 membranes-13-00359-f013:**
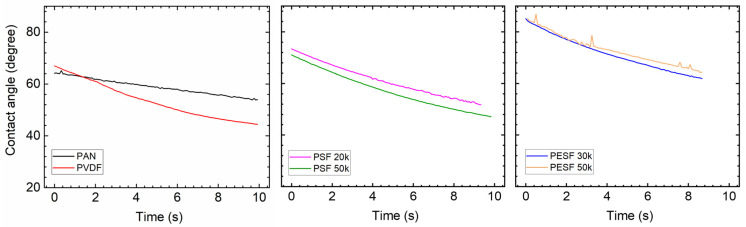
Contact angle of different porous support coated with PVA solution.

**Figure 14 membranes-13-00359-f014:**
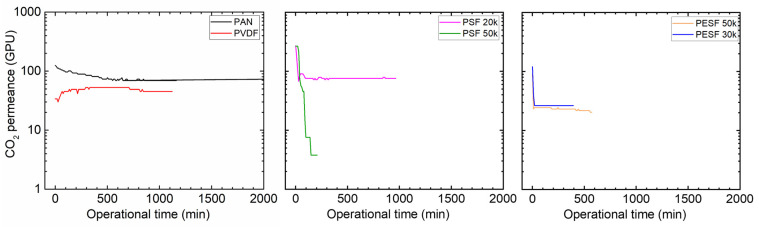
CO_2_ permeance of PVA thin-film composite membrane coated on 6 selected supports with a feed pressure of 2 bar at 35 °C, 100%RH.

**Table 1 membranes-13-00359-t001:** The overall thickness of the 6 porous supports selected.

Support	Pore Type	Top Layer Thickness (μm)	Total Thickness (μm)
PAN	Sponge	--	~40
PESF 30k	Finger	~6	~75
PESF 50k	Finger	~18	~80
PSF 20k	Finger	5~20	~150
PSF 50k	Sponge	--	120~150
PVDF	Sponge + Finger	~1	70~85

**Table 2 membranes-13-00359-t002:** The selective layer thickness of the Pebax TFC membrane fabricated on various supports.

Supports	Thickness (nm)
PAN	193.5 ± 18.3
PVDF	325.5 ± 23.4
PSF 20k	366.7 ± 50.9
PSF 50k	142.8 ± 36.5
PESF 30k	815.2 ± 70.0
PESF 50k	1180.2 ± 86.5

**Table 3 membranes-13-00359-t003:** The selective layer thicknesses of PVA TFC membranes fabricated on different supports.

Supports	Thickness (nm)
PAN	325 ± 36
PVDF	236 ± 11.7
PSF 20k	114.2 ± 53.8
PSF 50k	175.1 ± 10
PESF 30k	350.8 ± 41.0
PESF 50k	693.5 ± 110.2

## Data Availability

The data presented in this study are available in insert article.

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
