# Peer review of "Effects of Porous Supports in Thin-Film Composite Membranes on CO2 Separation Performances"

_membranes, 2023, doi:10.3390/membranes13030359_

Round 1

Reviewer 1 Report

The authors present a study of potentially great interest for the application of multilayered thin film composite membranes to the separation of CO2 from various, humidified gas streams. The study concentrates on the impact of the mass transfer resistance imposed by porous support layers on the overall permeation performance should condensation be allowed to occur in the support. Therein lies the main deficiency of this study. The experiments appear to be devised to trigger this effect whilst these conditions will hardly be achieved in reality. The authors should clearly state this in a revised version and point out why think these conditions are relevant. They also should explain why their findings were not observed in various long term studies (e.g. Han et al. Journal of Membrane Science 575 (2019) 242–251, Hägg et al. Energy Procedia 114 ( 2017 ) 6150 – 6165, Brinkmann et al. Engineering 3 (2017) 485–493) using membranes similar in structure and superior in permeation performance compared to those used in this study.

In essence, the study should only be published if revised according to the points raised above.

The following specific points should be addressed:

Page 1, line 27: Whilst PVA is known for its selectivity to water, it is not well suited to be employed as a material for a gas separation membrane. Why was it selected? The transport mechanism in both materials is purely solution-diffusion, i.e. not facilitated transport. Why was no facilitated transport membrane investigated?

Section 1 Introduction: Please refrain from quoting how often a paper was cited. A specific study is either relevant to the work presented or not. The amount of citations should not have an impact.

Page 2, line 44: Include amount of between enormous and new.

Page 2, lines 47-48: Whilst multilayer composite membranes do play an important role, e.g. in membranes employed for organic vapour recovery, the majority of gas separation membranes employed in industry are integral asymmetric hollow fibre membranes.

Page 3, lines 110-112: This logic is not clear. A gutter layer often is necessary to make the application of a thin separation layer possible, i.e. increasing the permeance of the membrane and hence the impact of the porous support. This would be good for this study.

Page 3, lines 120-122: The authors should give reasons for the use of a sweep gas assisted system as opposed to the more common set up of a 3 end system, i.e. using the pressure difference only to generate the fugacity driving force. Furthermore, the set-up  as e.g. given by Dai et al in [23] should be shown to help the reader.

Page 3, lines 130-131: Since pressures are quite low, real gas behaviour is not likely to have a large impact on the result. However, was this impact checked to be negligible?

Page 5, lines 175-179: These sentences are not clear: were all supports but Psf used as received or were they washed before being coated?

Page 7, lines 199-200: There are no error bars in Fig. 5.

Page 7, line 218:   This needs to be clarified: does fully humid mean that both feed and purge gas streams were fully humidified?

Page 8, lines 226-238: The argumentation is sound - but only valid for a system as described here. In typical operation scenarios involving gas separation systems without purge gas the conditions for pore condensation are typically not reached. The authors should state this clearly and add tables indicating he variable ranges investigated including the required conditions for pore condensation.

Page 8, lines 247-256: The authors repeatedly refer to facilitated transport membranes but in fact are using materials that work according to the solution diffusion mechanism. Why? The membrane presented is not competitive as a CO2/N2 separation membrane, see e.g. the Polaris membrane of MTR (e.g. Merkel et al. Journal of Membrane Science 359 (2010) 126–139), the facilitated transport membranes of Prof Ho's group at OSU (e.g. Han et al. Journal of Membrane Science 575 (2019) 242–251), the Membranes form NTNU (e.g. Hägg et al. Energy Procedia 114 ( 2017 ) 6150 – 6165, He Engineering 7 (2021) 124–131) or PolyActive TFCM (e.g. Brinkmann et al. Engineering 3 (2017) 485–493). These membranes exceed CO2 permeances of 1000 in mixed, humid gas. All of these membranes were tested under long term flue gas conditions and showed stable operating behaviour under prolonged periods of being exposed to humid flue gas were none of the effects treated in this study were observed. This does not mean that the current study is not worthwhile. The subject of the study has to be put into context of other studies that appear to contradict the results presented here.

Page 9, Figure 8: What support was used?  1000 min hardly qualifies as a time long enough for long term stability checks.

Page 9, line 266. The selectivity is ok, the permeance is too low - by an order of magnitude. Probably because no gutter layer was used.

Page 11, Figure 11: Where do the fluctuations for PAN/PEBAX come from?

Page 9: lines 323-324: This statement should be put into relation to the value measured for the dry gas mixtures. Is the initial 500 GPU for PEBAX/PAN as opposed to the initial value of 150 GPU for the dry mixture attributed to the humidity effect?

Page 15: lines 425-427: This is only true for the conditions selected in this study. The generalisation drawn is not valid.

Author Response

Dear editor,

The authors would like to thank the reviewers and the editor for their valuable comments and suggestions to help us improve this manuscript. We have carefully considered all the reviewers’ comments and revised the manuscript to address their concerns. To aid in the reviewing process, we have replied to all the comments on a point-by-point basis marked green and highlighted the revised sections of the main manuscript marked as red. We hope the manuscript can now be accepted for publishing in Membranes.

Thank you very much for your kind work!

Sincerely,

Zhongde Dai

Reviewer 2 Report

Dear authors,

Your manuscript investigated the effect of porous supports on TFC membrane performances when water is involved. 

Your study is quite interesting, as the support is playing an important role in the membrane performance and few studies have been done on it.

I am happy to recommend your manuscript for publication after minor revisions.

My comments are:

 -  some errors in text editing: p3, L99;p3 L115; p8 L244

- p3 L115-117: the sentence is unclear

- on the plots, you should use one color per support ( so 6 colors)

- you should indicate the full name of the polymers you used in the Materials part: PSf, PESf, PAN, ...

- why are you using CH4 as a sweep gas? The permeability of CH4 is higher than N2 for PEBAX...You might have some interference..

- Figure 5 : the error bars are missing

- Figure 8: which support has been used?

- I would put only one table with the thickness after coating with one column for PEBAX and one for PVA. It will be easy to compare the thickness. Could you relate the thickness to the CA of the support?

- What is the value of CA of a self standing PEBAX and PVA? These values should be compared to teh CA you measured after coating, and it might be a good indication to check if your coating is defect-free.

- Gas permeation over time: the authors should indicate why the operation time is not the same for all the membranes.

Author Response

Please find the detailed respond in the attachment.

Reviewer 3 Report

The manuscript reports the effects of commonly used porous supports on membrane morphology and gas transport properties of TFC membranes. It was found that porous supports contribute significantly to the overall mass transfer resistance, and the presence of water vapor worsens the mass transfer in the porous support due to pore condensation and support material swelling. Given the importance of the topics on membrane gas separations, these results have broad interests. The manuscript is well-written, and the technical content is clearly communicated to the readership. I would suggest publication in Membranes.

Author Response

Thank you for your kind comment.

Round 2

Reviewer 1 Report

The points mentioned in my original review have been addressed. I still feel that the paper is sound in its setup - but only relevant in an academic context.